# Impact of Long-Term Statin Therapy on Incidence and Severity of Community-Acquired Pneumonia: A Real-World Data Analysis

**DOI:** 10.3390/biomedicines13061438

**Published:** 2025-06-11

**Authors:** Diana Toledo, Àurea Cartanyà-Hueso, Rosa Morros, Maria Giner-Soriano, Àngela Domínguez, Carles Vilaplana-Carnerero, María Grau

**Affiliations:** 1Department of Medicine, School of Medicine and Health Sciences, University of Barcelona, 08036 Barcelona, Spainangela.dominguez@ub.edu (À.D.); carles_vilaplana@ub.edu (C.V.-C.); 2Biomedical Research Consortium in Epidemiology and Public Health (CIBERESP), 28029 Madrid, Spain; 3Fundació Institut Universitari per a la Recerca a l’Atenció Primària de Salut Jordi Gol i Gurina (IDIAPJGol), 08007 Barcelona, Spain; acartanya@idiapjgol.org (À.C.-H.); rmorros@idiapjgol.org (R.M.); mginer@idiapjgol.info (M.G.-S.); 4Biomedical Research Consortium in Infectious Diseases (CIBERINFEC), 28029 Madrid, Spain; 5Department of Pharmacology, Therapeutics and Toxicology, School of Medicine, Universitat Autònoma de Barcelona, 08193 Bellaterra, Spain; 6School of Medicine, Universitat Autònoma de Barcelona, 08193 Bellaterra, Spain; 7Service for the Promotion of Quality and Bioethics, General Directorate of Health Planning and Regulation, Department of Health, Government of Catalonia, 08028 Barcelona, Spain

**Keywords:** Hydroxymethylglutaryl-CoA reductase inhibitors, community-acquired pneumonia, electronic health records, cohort studies, pharmacoepidemiology

## Abstract

**Objectives:** This study aims to evaluate the impact of chronic statin therapy on the incidence of community-acquired pneumonia (CAP) and the rate of intensive care unit (ICU) admissions associated with CAP. **Methods**: Two population-based dynamic cohorts, consisting of individuals exposed and unexposed to statins, were followed from 2010 to 2019. Participants were older than 60 years, with frail patients excluded. The primary outcomes were the incidence of CAP and ICU admissions due to CAP, serving as a proxy for complicated cases. The exposed cohort included new statin users with at least two pharmacy invoices within 90 days of the recruitment period. Adjusted risk ratios (aRRs) for CAP incidence and CAP-associated ICU admissions were calculated using Poisson regression. **Results**: This study analyzed a sample of 639,564 individuals, evenly divided into exposed (319,782) and unexposed (319,782) groups, with a mean age of 71 years (standard deviation of 8 years) and 57% women. New statin users had a higher incidence of CAP [42.1 (95% confidence interval: 41.9–42.2) vs. 36.6 (36.5–36.8) per 1000 person-years] and ICU admissions [11.5 (11.5–11.6) vs. 10.1 (10.0–10.1) per 1000 person-years] compared to non-users. The adjusted analysis indicated that statin treatment reduced CAP risk by 6% [aRR: 0.94 (0.91–0.96)] and ICU admission by 7% [aRR: 0.93 (0.88–0.98)]. **Conclusions**: Prior statin therapy was associated with a clinically significant reduction in the incidence of CAP and ICU admissions due to CAP, despite the greater vulnerability of new users at the start of treatment compared to non-users.

## 1. Introduction

Community-acquired pneumonia (CAP) is a leading cause of death from infectious diseases globally and a significant public health concern in Europe [1]. The annual incidence of CAP ranges from 1.07 to 1.20 per 1000 person-years, escalating to 14 per 1000 person-years among adults aged 65 or older [2]. CAP is a major driver of hospitalizations, with approximately one million people admitted annually in Europe, and up to 50% of cases necessitating inpatient care [3,4].

Statins, 3-hydroxy-3-methylglutaryl-coenzyme-A inhibitors, are well-known lipid-lowering agents and have been clinically applied in the prevention of cardiovascular diseases [5]. In addition, a large number of studies have shown the pleiotropic effects of statins beyond cholesterol-lowering action [6,7,8]. Thus, statins have shown anti-inflammatory activity in healthy subjects, reducing TNF-α, CRP, and metalloproteinase levels after lipopolysaccharide exposure [9]. Additionally, they appear to enhance the humoral immune response to the administration of the 23-valent pneumococcal polysaccharide vaccine [10,11,12]. Indeed, the lung-specific association of statins has been thoroughly investigated. While some clinical studies have demonstrated a protective effect from chronic statin treatment [13,14], others have found no significant impact [15,16].

The analysis of real-world data collected from electronic health record (EHR) databases can help shed light on the association between chronic statin treatment and the risk and severity of CAP. These databases contain lifelong health information, include a large number of individuals with diverse risk profiles, provide long-term follow-up, and encompass a variety of outcomes, such as CAP diagnosis and intensive care unit (ICU) admissions [17]. Thesee databases have previously supported research on both the epidemiology of pneumonia [18] and the anti-inflammatory effects of statins [19]. By leveraging this comprehensive dataset, we aim to provide additional insights into the potential association between chronic statin therapy and pneumonia incidence, thereby addressing an important and relevant clinical question.

The objective of this study was to evaluate the impact of prior use of chronic statin therapy on the incidence of CAP, as well as the rate of ICU admissions associated with this condition.

## 2. Materials and Methods

### 2.1. Study Design and Data Source

Two population-based dynamic cohorts (exposed and non-exposed to statins) were followed from 2010 to 2019 using the Information System for Research in Primary Care (SIDIAP from the Spanish term Sistema de Información para el Desarrollo de la Investigación en Atención Primaria).

The SIDIAP database comprises pseudonymized longitudinal patient records routinely collected by over 30,000 professionals from the Catalan Institute of Health. This comprehensive database includes records for 6 million individuals, representing 80% of the Catalan population and 10.2% of Spain’s population. The recorded information encompasses demographic and lifestyle factors, clinical diagnoses, outcomes, events, referrals, and hospital discharges coded by the International Classification of Diseases, Tenth Revision (ICD-10); laboratory tests; and prescribed medications dispensed by community pharmacies [17]. The study protocol was approved by the local ethics committees of the two involved institutions (e.g., IDIAP Jordi Gol [23/094-EOM] and the University of Barcelona [IRB00003099]).

### 2.2. Participants and Exposure Assessment

Participants were enrolled in the cohort upon reaching 60 years of age, or on 1 January 2010, if they were already 60 years old at the start of recruitment. Frail patients, defined as those diagnosed with cancer, dementia, paralysis, or organ transplant, as well as those undergoing dialysis or those institutionalized, were excluded.

Statin exposure was defined as the use of a drug classified under the Anatomical Therapeutic Chemical (ATC) Classification System code C10AA. This includes ATC C10 drugs containing, either as monotherapy or in combination, ATC C10AA01 to 05 (simvastatin, lovastatin, pravastatin, fluvastatin, and atorvastatin) and C10AA07 (rosuvastatin). The exposure cohort consisted of new statin users, identified from database records, with at least two pharmacy invoices within 90 days of the recruitment period. The index date was defined as the first statin billing. These patients must not have been exposed to statins prior to entering the cohort (billing period 2006–2010). Users of cholesterol-lowering drugs other than statins (different from ATC C10AA), including those with ATC C10AB, C10AC, C10AD, and C10AX not combined with a C10AA, and those with fewer than two statin billings or more than 90 days between the first and second billing were excluded from the exposure cohort. Non-users, or those not exposed to statins, were defined as individuals with at least one visit to their healthcare provider within the 18 months prior to the index date of the matched individual.

### 2.3. Outcomes

The SIDIAP codes for CAP were identified in both primary care and hospital discharge records. The outcomes considered at the follow-ups were as follows: (1) incidence of CAP, and (2) ICU admission associated with CAP as a proxy for complicated cases.

### 2.4. Baseline Covariates

The baseline period was defined as 2 years before the index date. The following covariates which may have influenced prescription decisions and study outcomes were considered: age, sex, MEDEA index [20], Charlson Comorbidity Index, dichotomous (yes/no) variables for diabetes or a record of antidiabetic drug use, hypertension or a record of antihypertensive drug use, dyslipidemia, a BMI > 30 kg/m^2^ (obesity), and pneumococcal vaccination. Smoking was summarized in three categories (current, former, and never a smoker).

### 2.5. Statistical Analysis

Propensity score matching was used to assess the association between statin use and CAP incidence or ICU admission. We employed 1:1 nearest neighbor propensity score matching without replacement, with the propensity score estimated using logistic regression of statin use on year of birth, sex, health center deprivation index (developed and validated for Spain by the MEDEA study) [20], tobacco status, and entry and exit to the cohort. We restricted the caliper to 0.1 and matched exactly with year of birth, and entry and exit to the cohort [21]. The adequacy of the matching procedure was assessed using standardized mean differences (SMDs), which quantify the difference in the means of each covariate between treatment groups, standardized by a common factor—typically the pooled standard deviation across both groups (e.g., treated and untreated). SMD values close to zero indicate good covariate balance [22].

Matching was conducted using the functions implemented in the MatchIt R package [23].

Sociodemographic and clinical characteristics were described according to statin exposure using absolute frequency and percentage for categorical variables and mean and standard deviation (SD) or median and first and third quartiles (IQR: interquartile range) for continuous variables. Differences in sociodemographic and clinical characteristics according to statin exposure were tested using the Chi-square test for the categorical variables and the T-Student or Wilcoxon rank sum test for the continuous variables. The incidence rate per 1000 people and its 95% confidence interval (CI) of CAP and CAP-associated ICU admission were calculated for those exposed and unexposed. We used the generalized linear model (glm) function implemented in R to fit the outcome model, and the avg_comparison function from the marginaleffects R package to perform g-computation on the matched sample, estimating the unadjusted and adjusted risk ratios and their 95% CIs for CAP and CAP-associated ICU admission. Cluster-robust variance estimation was applied to calculate standard errors, using matching stratum membership as the clustering variable [24]. Cluster-robust variance estimation was applied to calculate standard errors, using matching stratum membership as the clustering variable. The adjustment variables were patient complexity (Charlson and deprivation index) and pneumococcal vaccination. In addition, we plotted cumulative incidence curves with their 95% CI of CAP and CAP-associated ICU admission with death as a competing risk using ggsurvfit in the ggsurvfit R package. Furthermore, we assessed the differences in CAP incidence and CAP severity survival distributions between statin users and statin non-users using the log-rank test (statistical significant level 0.05) [25]. Calculations were made with the R statistical package (R Core Team; version 4.4.3) [26].

## 3. Results

### 3.1. Recruitment

The initial sample, recruited between 1 January 2010 and 31 December 2019, included 1,779,709 individuals. After exclusions, the exposed cohort included 394,014 new statin users, while the unexposed cohort comprised 773,713 individuals. Propensity score matching was applied, resulting in a final sample of 639,564 individuals, with 319,782 in the exposed group and 319,782 in the unexposed group (Figure 1). Appendix A presents the SMDs for all covariates included in the matching procedure.

Median follow-up was 8.9 years (5.2, first quartile; 9.9, third quartile). Women constituted 56.7% of the study population and the mean age was 71 (SD = 8) years. Diabetes was present in nearly 10.3% of the participants, hypertension in 33.5%, smoking in 11.4%, and dyslipidemia in 24.9%. The baseline characteristics for new users and non-users are presented in Table 1.

### 3.2. Outcomes According the Use of Statins

From 2010 to 2019, new statin users had a higher incidence of CAP compared to non-users, with rates of 42.1 per 1000 person-years (95% CI: 41.9–42.2) versus 36.6 per 1000 person-years (95% CI: 36.5–36.8). Statin users also experienced more severe cases requiring ICU admission, with rates of 11.5 per 1000 person-years (95% CI: 11.5–11.6) compared to 10.1 per 1000 person-years (95% CI: 10.0–10.1) among non-users. Figure 2 and Figure 3 present the cumulative incidence curves and log-rank test for the incidence and severity of CAP, indicating statin treatment reduced ICU admission by 7% [adjusted RR: 0.93 (95% CI: 0.88–0.98)] (Table 2).

The adjusted analysis incorporating additional indicators of patient complexity (i.e., Charlson Comorbidity Index) and the pneumococcal vaccination status indicated that statin treatment reduced adjusted CAP risk by 6% [adjusted RR: 0.94 (95% CI: 0.91–0.96)]. Appendix A presents the results of the progressively complex models, each incorporating additional covariates to assess the robustness of the findings.

## 4. Discussion

Despite the increased vulnerability of individuals on statin treatment, these patients exhibited a lower incidence of CAP and reduced severity, as indicated by fewer ICU admissions, compared to those not receiving statin therapy. These findings underscore the pleiotropic effects of statins, which extend beyond their primary role in lipid-lowering. Statins possess potential anti-inflammatory properties that may play a crucial role in mitigating the risk and severity of CAP. This anti-inflammatory activity could help reduce vulnerability to infections and improve outcomes for patients on chronic statin therapy. Furthermore, the observed benefits in reducing ICU admissions and overall disease severity highlight the importance of considering statin therapy as a multifaceted approach to managing patients at risk for CAP.

### 4.1. Potential Lung-Protective Effect of Statins

Our findings, consistent with several previous population-based case-control studies, suggest a protective association between prior statin use and pneumonia outcomes [27,28,29]. However, it is important to acknowledge that retrospective studies are inherently susceptible to confounding. Dublin et al. conducted a population-based case-control study that accounted for multiple confounders and reported an increased risk of CAP among statin users [30]. In contrast, Nielsen et al. demonstrated that, after adjustment for the relevant covariates, the association between statin use and CAP-related hospitalizations shifted from a harmful to a beneficial effect—findings that align closely with our results regarding both the incidence and severity of pneumonia [31]. These contrasting outcomes underscore the importance of rigorous adjustment for baseline differences, as unadjusted analyses may reflect underlying health disparities rather than true treatment effects. Our adjusted analysis, therefore, offers a more accurate estimate of the potential protective role of statins in pneumonia.

Statins might bolster host defense against infection [9], given that this effect is associated most consistently with a reduced risk of incident pneumonia [32]. Coston et al. have shown that preadmission statin use was associated with a lower risk of pneumonia than the other clinical presentations of a *Burkholderia pseudomallei* infection (melioidosis) [33]. Preclinical studies have demonstrated that statins can bolster the body’s defense against pneumonia through various mechanisms such as enhancing the recruitment of inflammatory cells to the lungs and improving bacterial clearance in a *Chlamydia pneumoniae* model [34]; reducing the expression of platelet-activating factor receptors on lung epithelial cells, thereby decreasing bacterial invasion in pneumococcal pneumonia [35,36]; protecting resident alveolar macrophages from pneumolysin-induced cell lysis, which aids in the clearance of *Streptococcus pneumoniae* [35]; and modulating the expression of the genes involved in host defense against *Pseudomonas aeruginosa* in lung epithelial cells [37]. Additionally, statin therapy may influence neutrophil recruitment and function in a way that reduces excessive pulmonary inflammation while still preserving the activity of protective proinflammatory mediators and the ability to kill bacteria [35].

### 4.2. Real World Data

The use of epidemiological research methodology to analyze EHR databases is particularly suitable for investigating the effects of chronic statin treatment on the incidence and severity of CAP [30,38]. In this study, we have specifically, gathered information to study the potential effects of exposure (statin treatment) on the outcomes (incidence and severity of CAP) through healthcare variables (diagnosis and ICU admission) and other modulating variables (e.g., sociodemographic information, comorbidities, or vaccination records). The characteristics of the National Health System in Spain make EHR databases (real-world data) a highly valuable source of information for epidemiological research, given their extensive population representativeness. They also offer the possibility of cross-referencing information at the individual level, allowing for the reconstruction of an individual’s various contacts with the healthcare system (primary and hospital care) [17].

Recently, several studies utilizing real-world data analysis have been conducted on this topic, particularly focusing on selected populations. Notably, Franchi et al. demonstrated that the combination of either of these two antihypertensive medications—angiotensin-converting enzyme inhibitor (ACE-I) or angiotensin II receptor blockers (ARBs) —with statins significantly reduced the risk of pneumonia in older adults. However, this risk reduction was not observed when the patients were treated with statins alone [39]. A real-world data study conducted in Southern California showed that statin use was associated with lower mortality in patients hospitalized with sepsis [40]. Finally, Sun et al. demonstrated that the persistent use of statins reduced sepsis and septic shock risk in patients with type 2 diabetes mellitus [41].

### 4.3. Characteristics and Limitations

A major strength of our study is the large sample of individuals drawn from a high-quality, internally validated database of EHR. This database provides high external validity and includes clinical data from patients that are often excluded from trials, such as women, older adults, and individuals with diabetes. However, several general limitations are inherent to observational studies using medical records. First, the potency of statins was not considered in the analysis, although a recent publication showed no significant association between statin potency and hospitalization for CAP [42]. Second, residual confounding is possible, particularly by indication. To avoid frailty bias, we excluded individuals with cancer, dementia, paralysis, organ transplant, those undergoing dialysis, or those institutionalized. We employed a new user design to minimize the potential effect of statins on confounding factors, matched for a propensity score. Third, nonclinical factors that may influence prescription patterns and treatment adherence were not measured which could lead to confounding and/or interaction bias. These factors include a doctor’s perception of a patient’s risk profile, prescriber experiences, unreported side effects, and patient perceptions of risks and willingness to take the drug [43]. Fourth, we cannot rule out selection bias because we only considered the user population of primary care services, as well as observer bias because the database consists of EHR. Fifth, influenza vaccination was not included in the adjusted models. Given that influenza vaccines are administered each season and individuals may receive them inconsistently across different seasons, accurately capturing vaccination status over time is challenging. This inconsistency complicates the assessment of cumulative exposure and the alignment of vaccination timing with pneumonia episodes. Including such a variable without precise temporal data could introduce misclassification bias and reduce the validity of the adjusted estimates and therefore, we opted to exclude influenza vaccination from the current analysis. Sixth, antimicrobial treatment was not included in the adjusted models. While the role of antimicrobial therapy in pneumonia management is indisputable, incorporating this variable into the models would have introduced substantial complexity. This is due to the wide variability in antimicrobial agents, dosages, timing of administration, and their indication based on patients’ baseline clinical status. Such heterogeneity could lead to confounding and reduce the interpretability of the model. Seventh, nearly 70% of the patients treated with statins did not have a formally coded diagnosis of hypercholesterolemia in the database. This discrepancy likely reflects under-recording or incomplete diagnostic coding in routine clinical practice, a common limitation in real-world data sources. It highlights the gap between clinical decision-making and administrative documentation, where treatment may be based on laboratory results or cardiovascular risk assessments not fully captured in diagnostic codes. Last, the causes of death are not available in the SIDIAP database, which precluded an analysis of statins’ effect on CAP mortality.

## 5. Conclusions

Prior statin therapy was associated with a potential reduction in the adjusted incidence of CAP and related ICU admissions, even when accounting for the increased vulnerability of new users at treatment initiation. These findings, derived from real-world data in electronic health records, suggest a possible beneficial role of statins in reducing the severity and perhaps the onset of CAP. However, these observational results should be interpreted with caution. Further validation through randomized controlled trials—both in patients with CAP and in populations at risk—is necessary to confirm these associations and to determine their clinical and public health significance.

## Figures and Tables

**Figure 1 biomedicines-13-01438-f001:**
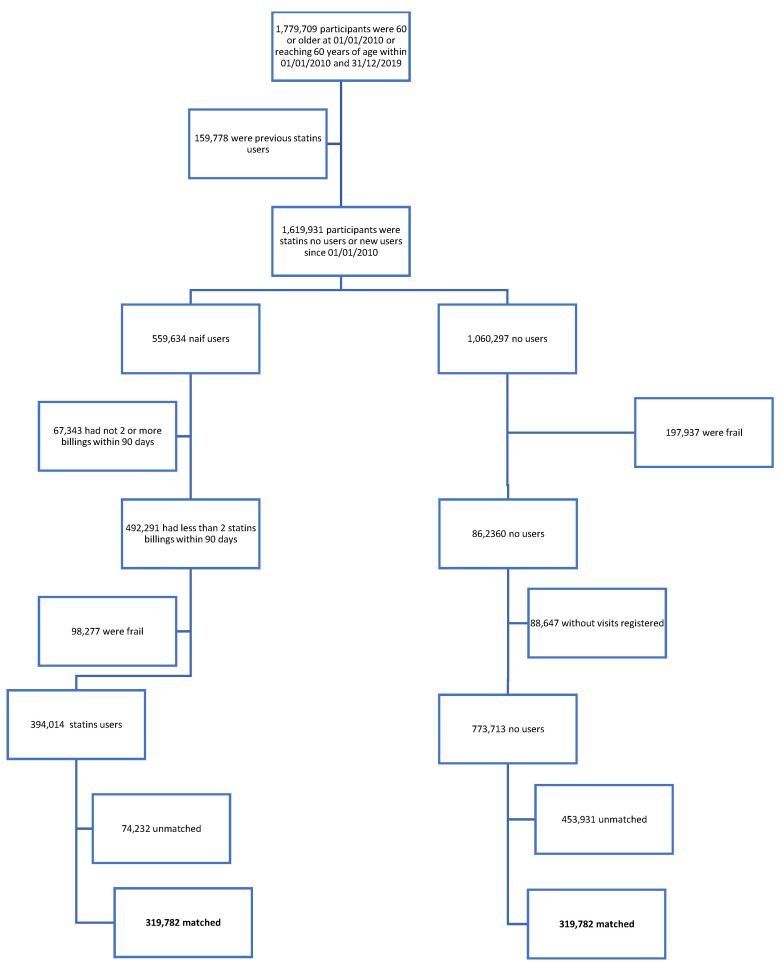
Flow-chart of participant selection.

**Figure 2 biomedicines-13-01438-f002:**
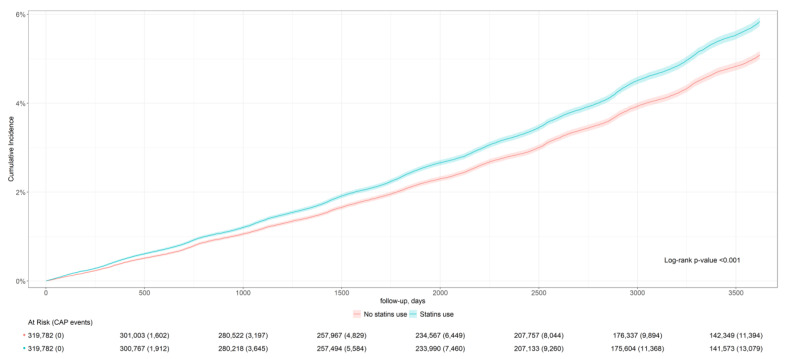
Cumulative incidence curve for ten-year CAP incidence.

**Figure 3 biomedicines-13-01438-f003:**
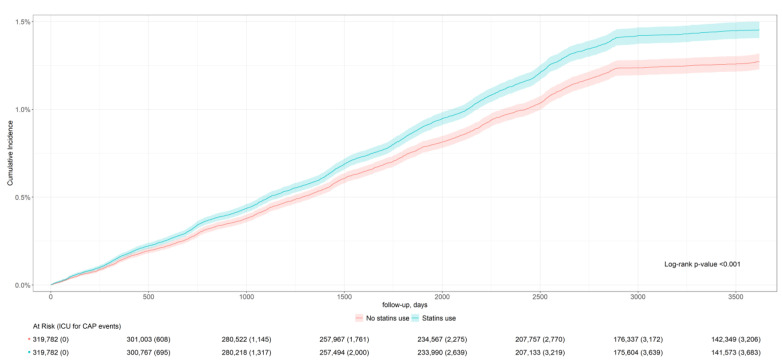
Cumulative incidence curve for ten-year ICU admission in CAP patients.

**Table 1 biomedicines-13-01438-t001:** Baseline characteristics of statin new-users and statin non-users.

	Statin New-Users N = 319,782	Statin Non-Users N = 319,782	*p*-Value	SMD
Sex, *n* (%) (women)	180,405 (56.4)	182,112 (56.9)	<0.001	0.011
Age, mean (SD)	71.09 (7.55)	71.09 (7.55)	0.963	<0.001
Smoking *n* (%)			<0.001	0.078
Smoker	33,532 (10.5)	39,519 (12.4)		
Former smoker	47,249 (14.8)	40,761 (12.7)		
Hypertension, *n* (%)	119,638 (37.4)	94,498 (29.6)	<0.001	0.167
Diabetes, *n* (%)	46,093 (14.4)	19,932 (6.2)	<0.001	0.271
Hypercholesterolemia, *n* (%)	98,696 (30.9)	60,454 (18.9)	<0.001	0.279
Obesity, *n* (%)	44,892 (14.0)	30,983 (9.7)	<0.001	0.135
Charlson index, median [IQR]	2.00 [1.00, 3.00]	2.00 [1.00, 2.00]	<0.001	0.167
MEDEA deprivation index, *n* (%)			<0.001	0.072
Rural	21,629 (6.8)	23,662 (7.4)		
Semi-rural	17,726 (5.5)	18,944 (5.9)		
Urban with low deprivation	62,725 (19.6)	68,602 (21.5)		
Semi-urban	39,256 (12.3)	38,647 (12.1)		
Urban with medium-low deprivation	50,357 (15.7)	51,499 (16.1)		
Urban with medium-high deprivation	68,173 (21.3)	62,318 (19.5)		
Urban with high deprivation	59,478 (18.6)	55,542 (17.4)		
Pneumococcal vaccination, *n* (%)	24,734 (7.7)	21,021 (6.6)	<0.001	0.045

IQR: Interquartile range. SD: Standard deviation. SMD: Standard mean difference.

**Table 2 biomedicines-13-01438-t002:** Incidence rate of CAP and ICU admission per 1000 persons-year, in statin new-users and non-users, respectively.

	Statin New Users		Statin Non-Users		
	Events	Incidence Rate (95% CI)	Events	Incidence Rate (95% CI)	Adjusted Risk Ratio * (95% CI)
CAP incidence	13,453	42.1 (41.9; 42.2)	11,715	36.6 (36.5; 36.8)	0.94 (0.91; 0.96)
CAP severity (ICU admission)	3690	11.5 (11.5; 11.6)	3223	10.1 (10.0; 10.1)	0.93 (0.88; 0.98)

CAP: Community acquired pneumonia. CI: Confidence interval. ICU: Intensive care unit. * Model adjusted for Charlson index and pneumococcal vaccine.

## Data Availability

Data are unavailable due to privacy and ethical restrictions.

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
