# Peer review of "Impact of Long-Term Statin Therapy on Incidence and Severity of Community-Acquired Pneumonia: A Real-World Data Analysis"

_biomedicines, 2025, doi:10.3390/biomedicines13061438_

Round 1

Reviewer 1 Report

Comments and Suggestions for Authors

Old question:
that has been debated for over a decade.

Strength: longitudinal
Database with good-quality follow-up. Large database.

Limitations:

Adequacy of
Matching process: very poor, with 2% more smokers in the unexposed group, for example.
patients!

Limitations:
The confounders not taken into account:

To improve:
To improve the manuscript, I suggest:

-
 The influenza vaccine should also be included in
the model. If it is unavailable, explain why and add this to the limitations.

-
In
The matched population's major residual confounders should be included in the
Final model of the multivariate analysis (e.g. smoking status, diabetes, obesity). At
From a statistical point of view, adjustment for the factors used for the propensity score
design is feasible and should be implemented here.

Discussion:
Many other large articles on statins and sepsis have been published in major
journals. To improve the manuscript, please include a table summarising the
most important studies and potential limitations of the available results.

Other suggestions:

-
The
Antimicrobial consumption in the exposed and unexposed populations is available in the
The database and should be reported in both groups.

Minor:
Table 1, add the SMD of each variable. It is far more important to be
It is important to be comfortable with the matching process.

Author Response

Old question: that has been debated for over a decade.

Reply: We appreciate the reviewer’s comment regarding the timeliness of our research topic. Indeed, the investigation of the pleiotropic effects of statins—particularly their potential role in modulating inflammatory responses—has been ongoing for several years. Pneumonia, as a disease with a significant inflammatory component, has been a focal point in this context. While some studies have reported a protective association between statin use and pneumonia outcomes (Chung SD. Clin. Microbiol. Infect. 2014; Grudzinska, FS. Clin. Med. (Lond). 2017), others have found no significant effect, highlighting the ongoing controversy and the need for further clarification (Zeenny RM. J Int Med Res. 2020; Batais, MA. Curr Infect Dis Rep. 2017).

Our study aims to contribute to this body of evidence by conducting a robust, confirmatory analysis using real-world data derived from electronic health records. The database we utilize has previously supported research on both the epidemiology of pneumonia (e.g., Ochoa-Gondar O. BMC Pulm Med. 2023) and the anti-inflammatory effects of statins (e.g., Garcia-Gil M. J Cardiovasc Pharmacol Ther. 2019). By leveraging this comprehensive dataset, we aim to provide additional insights into the potential association between chronic statin therapy and pneumonia incidence, thereby addressing an important and still-relevant clinical question.

We have improved this view in the new version of the manuscript (Introduction, lines 59-68): “The analysis of real-world data collected from electronic health record (EHR) databases can help shed light on the association between chronic statin treatment and the risk and severity of CAP. These databases contain lifelong health information, include a large number of individuals with diverse risk profiles, provide long-term follow-up, and encompass a variety of outcomes, such as CAP diagnosis and intensive care unit (ICU) admissions (Recalde M. Int J Epidemiol. 2022). This database has previously supported research on both the epidemiology of pneumonia (Ochoa-Gondar O. BMC Pulm Med. 2023) and the anti-inflammatory effects of statins (Garcia-Gil M. J Cardiovasc Pharmacol Ther. 2019). By leveraging this comprehensive dataset, we aim to provide additional insights into the potential association between chronic statin therapy and pneumonia incidence, there-by addressing an important and relevant clinical question”.

We have added three new citations to Introduction:

#12 Rombauts, A.; Abelenda-Alonso, G.; Cuervo, G.; Gudiol, C.; Carratalà, J. Role of the inflammatory response in communi-ty-acquired pneumonia: clinical implications. Expert Rev. Anti. Infect. Ther. 2022, 20, 1261–1274. doi: 10.1080/14787210.2021.1834848

#18 Ochoa-Gondar, O.; Torras-Vives, V.; de Diego-Cabanes, C.; Satué-Gracia, E.M.; Vila-Rovira, A.; Forcadell-Perisa, M.J.; Ribas-Seguí, D.; Rodríguez-Casado, C.; Vila-Córcoles, A. Incidence and risk factors of pneumococcal pneumonia in adults: a population-based study. BMC Pulm Med. 2023, 23, 200. doi: 10.1186/s12890-023-02497-2.

#19 Garcia-Gil, M.; Comas-Cufí, M.; Ramos, R.; Martí, R.; Alves-Cabratosa, L.; Parramon, D.; Prieto-Alhambra, D.; Baena-Díez, J.M.; Salvador-González, B.; Elosua, R.; Dégano, I.R.; Marrugat, J.; Grau, M. Effectiveness of statins as primary prevention in people with gout: A population-based cohort study. J. Cardiovasc. Pharmacol. Ther. 2019, 24, 542–550. doi: 10.1177/1074248419857071

Strength: longitudinal database with good-quality follow-up. Large database.

Limitations: Adequacy of matching process: very poor, with 2% more smokers in the unexposed group, for example. patients!

Reply: Regarding the adequacy of the matching process, we have included a new Figure (Supplementary Figure 1 in the new version of the manuscript) that displays the Standardized Mean Differences (SMDs) for all covariates used in the matching procedure. In this figure, red dots represent the SMDs before matching, and blue dots represent the SMDs after matching. According to the literature, a matching process is generally considered adequate when post-matching SMDs fall within the range of –0.1 to 0.1, with values closer to zero indicating better balance (Ali MS et al., Pharmacoepidemiol Drug Saf. 2014). This threshold is visually represented by the dashed horizontal lines in the figure. As all blue dots fall within this acceptable range, we consider the matching process to be appropriate and effective in balancing covariates between groups.

In addition, a sentence has been added in the new version of the manuscript (Results, lines 160-161): “Supplementary Figure 1 presents the SMDs for all covariates included in the matching procedure”.

Supplementary Figure 1. Standardized Mean Differences (SMDs) for all covariates before and after matching.

Limitations: The confounders not taken into account:

Reply: To address this, we have included a new Supplementary Table 1, which presents results from a series of increasingly complex models (Models 1–6), each incorporating additional covariates. This stepwise modeling approach allowed us to evaluate the impact of different sets of confounders on the outcome.

In line with the principle of parsimony, we selected Model 3 as the primary model for reporting our main results. This model includes a comprehensive yet balanced set of adjustment variables that capture the majority of variability in patient characteristics without introducing excessive complexity or multicollinearity. While Models 4–6 include additional variables such as detailed cardiovascular risk factors, we observed signs of overadjustment, which may obscure true associations and reduce model interpretability. For this reason, we opted not to use these models as the basis for our main findings. Instead, we chose to use the Charlson Comorbidity Index as a summary measure of patient complexity. This index is widely validated and captures a broad spectrum of comorbid conditions, offering a robust proxy for overall health status without fragmenting the model with numerous individual risk factors.

We have added a brief explanation of Supplementary Table 1 in the revised manuscript (lines 181–182): 'Supplementary Table 1 presents the results of progressively complex models, each incorporating additional covariates to assess the robustness of the findings.”      

Supplementary Table 1. Summary of Models 1–6 with Increasing Covariate Adjustment

Model 1

Model 2

Model 3

Model 4

Model 5

Model 6

Crude risk ratio

(95% CI)

Adjusted risk ratio

(95% CI)

Adjusted risk ratio

(95% CI)

Adjusted risk ratio

(95% CI)

Adjusted risk ratio

(95% CI)

Adjusted risk ratio

(95% CI)

CAP incidence

1.15  (1.12;1.18)

0.93 (0.91;0.96)

0.94 (0.91;0.96)

0.98 (0.95;1.01)

0.98 (0.95;1)

0.97 (0.95;1)

CAP severity

(ICU admission)

1.15 (1.09;1.2)

0.92 (0.87;0.97)

0.93 (0.88;0.98)

0.98 (0.93;1.03)

0.98 (0.93;1.03)

0.97 (0.92;1.02)

CAP: Community acquired pneumonia. CI: Confidence interval. ICU: Intensive care unit

Model 2 adjusted for Charlson index

Model 3 adjusted for Charlson index and pneumococcal vaccine

Model 4 adjusted for Charlson index, diabetes, hypercolesterolemia, obesity and hypertension

Model 5 adjusted for Charlson index, pneumococcal vaccine, diabetes, hypercolesterolemia, obesity and hypertension

Model 6 adjusted for Charlson index, pneumococcal vaccine, diabetes, hypercolesterolemia, obesity, hypertension and smoker

To improve: To improve the manuscript, I suggest:

- The influenza vaccine should also be included in the model. If it is unavailable, explain why and add this to the limitations.

Reply: We chose not to include this variable due to the complexity of accurately capturing participants’ vaccination status. Influenza vaccination is administered annually, and individuals may receive it inconsistently across years. As a result, it is challenging to assess cumulative exposure or to determine the timing and dosage relative to pneumonia episodes. This variability could introduce misclassification bias and limit the interpretability of the adjusted model. For these reasons, we opted to exclude influenza vaccination from the current analysis. We have added this point as a limitation in the new version of the manuscript (lines 271-278): “Fifth, influenza vaccination was not included in the adjusted models. Given that influenza vaccines are administered annually and individuals may receive them inconsistently across different seasons, accurately capturing vaccination status over time is challenging. This inconsistency complicates the assessment of cumulative exposure and the alignment of vaccination timing with pneumonia episodes. Including such a variable without precise temporal data could introduce misclassification bias and reduce the validity of the adjusted estimates. Therefore, we opted to exclude influenza vaccination from the current analysis.”

- In the matched population's major residual confounders should be included in the final model of the multivariate analysis (e.g. smoking status, diabetes, obesity). At from a statistical point of view, adjustment for the factors used for the propensity score design is feasible and should be implemented here.

Reply: To evaluate the adequacy of the matching process, we have included a new Supplementary Figure 1 in the revised manuscript. This figure illustrates the Standardized Mean Differences (SMDs) for all covariates used in the matching procedure. As shown, all post-matching SMDs (represented by blue dots) fall within the accepted threshold range, supporting the effectiveness of the matching in achieving balance across baseline characteristics.

To account for patient complexity in a robust yet parsimonious manner, we included the Charlson Comorbidity Index in the final multivariate model. This validated summary measure captures a broad spectrum of comorbid conditions and serves as a reliable proxy for overall health status. While individual cardiovascular risk factors such as smoking, diabetes, and obesity were considered, we found that the Charlson Index offered a more stable and interpretable adjustment without overcomplicating the model. We believe this approach strikes an appropriate balance between statistical rigor and model clarity, while still addressing key potential confounders. For transparency, we have also included all adjusted models in Supplementary Table 1.

Discussion:

Many other large articles on statins and sepsis have been published in major journals. To improve the manuscript, please include a table summarising the most important studies and potential limitations of the available results.

Reply: We appreciate the reviewer’s thoughtful suggestion regarding the inclusion of a summary table of previous publications. While our manuscript is specifically focused on evaluating the association between chronic statin use and both the incidence of pneumonia and ICU admissions using real-world data, we have thoroughly contextualized our findings within the existing literature in the Discussion section. There, we highlight key studies on this topic and discuss how our approach addresses several limitations identified in earlier research.

Nonetheless, we recognize the value of a comprehensive synthesis of prior evidence. We believe that developing a summary table, as proposed, would be best suited to a dedicated systematic review and meta-analysis. We consider this an excellent idea and one that we plan to explore in future work.

Other suggestions:

- The antimicrobial consumption in the exposed and unexposed populations is available in the database and should be reported in both groups.

Reply: Antimicrobial treatment was not included in the adjusted models. While the role of antimicrobial therapy in pneumonia management is indisputable, incorporating this variable into the models would have introduced substantial complexity. This is due to the wide variability in antimicrobial agents, dosages, timing of administration, and their indication based on patients’ baseline clinical status. Such heterogeneity could lead to confounding and reduce the interpretability of the model. Nonetheless, we acknowledge this as a relevant factor and have now included it as a limitation in the revised manuscript (lines 278-284): “Antimicrobial treatment was not included in the adjusted models. While the role of antimicrobial therapy in pneumonia management is indisputable, incorporating this variable into the models would have introduced substantial complexity. This is due to the wide variability in antimicrobial agents, dosages, timing of administration, and their indication based on patients’ baseline clinical status. Such heterogeneity could lead to confounding and reduce the interpretability of the model”.

Minor: Table 1, add the SMD of each variable. It is far more important to be it is important to be comfortable with the matching process.

Reply: We have updated Table 1 to include a new column presenting the Standardized Mean Differences (SMD) for each covariate. Additionally, we have expanded the Statistical Analysis section (lines 128–132) to include a definition and rationale for using SMDs: “The adequacy of the matching procedure was assessed using standardized mean differences (SMDs), which quantify the difference in the means of each covariate between treatment groups, standardized by a common factor—typically the pooled standard deviation across both groups (e.g., treated and untreated). SMD values close to zero indicate good covariate balance (Ali MS et al., Pharmacoepidemiol Drug Saf. 2014).”

We have added a new citation to Methods:

#22 Ali, M.S.; Groenwold, R.H.; Pestman, W.R.; Belitser, S.V.; Roes, K.C.; Hoes, A.W.; de Boer, A.; Klungel, O.H. Propensity score balance measures in pharmacoepidemiology: a simulation study. Pharmacoepidemiol. Drug Saf. 2014, 23, 802–811. doi.org: 10.1002/pds.3574

Reviewer 2 Report

Comments and Suggestions for Authors

Major remarks:

According to what is presented at the paragraph 3.2, Table 2, and Figures 2-3, statin users had higher CAP incidence and ICU admissions compared to non-users. Therefore, I do not understand why the autors conclude that statin treatment reduced CAP incidence and severity. Please, re-write this part of the article to become understandable for the readers.

Minor remarks:

  1. Line 92: correct the name of the 4th medication
  2. Table 1: what does it mean that hypercholesterolemia was observed in 30.9% of statin users? What was, therefore, the reason for statin introduction in the rest of statin-users? Please, correct.
  3. Figures 2-3 are unreadable. Please, correct.

Author Response

Reviewer #2

Major remarks:

According to what is presented at the paragraph 3.2, Table 2, and Figures 2-3, statin users had higher CAP incidence and ICU admissions compared to non-users. Therefore, I do not understand why the authors conclude that statin treatment reduced CAP incidence and severity. Please, re-write this part of the article to become understandable for the readers.

Reply: We appreciate the reviewer’s observation regarding the differences between the unadjusted and adjusted analyses of pneumonia outcomes. In the unadjusted analysis, individuals receiving statin therapy appeared to have worse outcomes. However, it is important to note that, despite matching on age, sex, and socioeconomic status, the statin-treated group exhibited a higher baseline cardiovascular risk profile, which likely influenced these initial results. To account for this imbalance, we conducted an adjusted analysis incorporating additional indicators of patient complexity, such as the Charlson Comorbidity Index, as well as pneumococcal vaccination status. After adjusting for these confounding factors, the association reversed—statin-treated individuals demonstrated better outcomes in terms of pneumonia incidence and severity. This suggests that the initial unadjusted findings were likely confounded by underlying health differences between the groups, and that the adjusted analysis provides a more accurate estimate of the potential protective effect of statins.

We have incorporated this explanation into the revised version of the manuscript to enhance clarity (lines 204–216): “Our findings, consistent with several previous population-based case-control studies, suggest a protective association between prior statin use and pneumonia out-comes (Vinogradova Y. Br J Gen. Pract. 2011; Myles PR. Pharmacoepidemiol Drug Saf. 2009; Schlienger RG. Pharmacotherapy. 2007). However, it is important to acknowledge that retrospective studies are inherently susceptible to confounding. For instance, Dublin et al. conducted a popula-tion-based case-control study that accounted for multiple confounders and reported an increased risk of community-acquired pneumonia (CAP) among statin users (Dublin S. BMJ. 2009). In contrast, Nielsen et al. demonstrated that, after adjustment for relevant covariates, the association between statin use and CAP-related hospitalizations shifted from a harm-ful to a beneficial effect—findings that align closely with our results regarding both the incidence and severity of pneumonia (Nielsen AG. Crit. Care. 2012). These contrasting outcomes underscore the importance of rigorous adjustment for baseline differences, as unadjusted analyses may reflect underlying health disparities rather than true treatment effects. Our ad-justed analysis, therefore, offers a more accurate estimate of the potential protective role of statins in pneumonia.”

Minor remarks:

Line 92: correct the name of the 4th medication

Reply: We have corrected the typo

Table 1: what does it mean that hypercholesterolemia was observed in 30.9% of statin users? What was, therefore, the reason for statin introduction in the rest of statin-users? Please, correct.

Reply: The variable “hypercholesterolemia” in Table 1 refers specifically to individuals with a recorded diagnosis of the condition in their medical history. However, it is important to note that not all patients treated with statins have a formally coded diagnosis of hypercholesterolemia in the database. This discrepancy likely reflects under-registration or incomplete coding of the diagnosis in routine clinical practice. In many cases, statins may have been prescribed based on elevated lipid levels, cardiovascular risk profiles, or secondary prevention indications, even if a formal diagnosis of hypercholesterolemia was not explicitly recorded. We have clarified this point in the revised manuscript to avoid confusion and better reflect the real-world nature of the data (lines 284-290): “Seventh, nearly 70% of patients treated with statins did not have a formally coded diagnosis of hypercholesterolemia in the database. This discrepancy likely reflects under-recording or incomplete diagnostic coding in routine clinical practice, a common limitation in real-world data sources. It highlights the gap between clinical decision-making and administrative documentation, where treatment may be based on laboratory results or cardiovascular risk assessments not fully captured in diagnostic codes”.

Figures 2-3 are unreadable. Please, correct

Reply: Figures 2 and 3 have been revised to enhance clarity and improve the visual presentation of the results.

Round 2

Reviewer 1 Report

Comments and Suggestions for Authors
  • The antimicrobial consumption in the exposed and unexposed populations is available in the database and should be reported in both groups.: NB: AB consumption reporting is a way to check that there is really a decrease in CAP and not only a masking of the diagnosis of CAP via an over use of antibiotics. 
  • I did not ask an adjustment but only a reporting on the antibiotic consumption...please add it (you can adjust the AB consumption without using it in the mulivariate model...)

Author Response

The antimicrobial consumption in the exposed and unexposed populations is available in the database and should be reported in both groups.: NB: AB consumption reporting is a way to check that there is really a decrease in CAP and not only a masking of the diagnosis of CAP via an over use of antibiotics.

I did not ask an adjustment but only a reporting on the antibiotic consumption...please add it (you can adjust the AB consumption without using it in the mulivariate model...)

Reply: We appreciate the reviewer’s comment and fully recognize the importance of reporting antimicrobial consumption in both exposed and unexposed populations. Although the SIDIAP database includes this information, we did not extract the variable for our analysis, as we had initially decided not to include it due to the complexity of interpreting antibiotic use—since it may reflect a wide range of clinical indications beyond pneumonia. Furthermore, assessing treatment adequacy or potential overtreatment is particularly challenging in the context of real-world data, where clinical decision-making is not always fully captured or standardized. 

Reviewer 2 Report

Comments and Suggestions for Authors

The paper was improved, thank you.

Comments on the Quality of English Language

Line 272: space is lacking in "seasonand"

Author Response

Line 272: space is lacking in "seasonand"

Reply: We have corrected the typo